# Real-time 3D reconstruction from single-photon lidar data using plug-and-play point cloud denoisers

Julián Tachella [1], Yoann Altmann[1]*, Nicolas Mellado[2], Aongus McCarthy [1], Rachael Tobin[1], Gerald S. Buller[1], Jean-Yves Tourneret[3] & Stephen McLaughlin [1]

Single-photon lidar has emerged as a prime candidate technology for depth imaging through challenging environments. Until now, a major limitation has been the significant amount of time required for the analysis of the recorded data. Here we show a new computational framework for real-time three-dimensional (3D) scene reconstruction from single-photon data. By combining statistical models with highly scalable computational tools from the computer graphics community, we demonstrate 3D reconstruction of complex outdoor scenes with processing times of the order of 20 ms, where the lidar data was acquired in broad daylight from distances up to 320 metres. The proposed method can handle an unknown number of surfaces in each pixel, allowing for target detection and imaging through cluttered scenes. This enables robust, real-time target reconstruction of complex moving scenes, paving the way for single-photon lidar at video rates for practical 3D imaging applications.

[1] School of Engineering and Physical Sciences, Heriot-Watt University, Edinburgh, UK. [2] IRIT, CNRS, University of Toulouse, Toulouse, France. [3] ENSEEIHT-IRIT-TeSA, University of Toulouse, Toulouse, France. *email: Y.Altmann@hw.ac.uk

Reconstruction of three-dimensional (3D) scenes has many important applications, such as autonomous navigation[1], environmental monitoring[2] and other computer vision tasks[3]. While geometric and reflectivity information can be acquired using many scanning modalities (e.g., RGB-D sensors[4], stereo imaging[5] or full waveform lidar[2]), single-photon systems have emerged in recent years as an excellent candidate technology. The time-correlated single-photon counting (TCSPC) lidar approach offers several advantages: the high sensitivity of single-photon detectors allows for the use of low-power, eye-safe laser sources; and the picosecond timing resolution enables excellent surface-to-surface resolution at long range (hundreds of metres to kilometres)[6]. Recently, the TCSPC technique has proved successful at reconstructing high resolution three-dimensional images in extreme environments such as through fog[7], with cluttered targets[8], in highly scattering underwater media[9], and in free-space at ranges greater than 10 km[6]. These applications have demonstrated the potential of the approach with relatively slowly scanned optical systems in the most challenging optical scenarios, and image reconstruction provided by post-processing of the data. However, recent advances in arrayed SPAD technology now allow rapid acquisition of data[10,11], meaning that full-field 3D image acquisition can be achieved at video rates, or higher, placing a severe bottleneck on the processing of data.

Even in the presence of a single surface per transverse pixel, robust 3D reconstruction of outdoor scenes is challenging due to the high ambient (solar) illumination and the low signal return from the scene. In these scenarios, existing approaches are either too slow or not robust enough and thus do not allow rapid analysis of dynamic scenes and subsequent automated decision-making processes. Existing computational imaging approaches can generally be divided into two families of methods. The first family assumes the presence of a single surface per observed pixel, which greatly simplifies the reconstruction problem as classical image reconstruction tools can be used to recover the range and reflectivity profiles. These algorithms address the 3D reconstruction by using some prior knowledge about these images. For instance, some approaches[12,13] propose a hierarchical Bayesian model and compute estimates using samples generated by appropriate Markov chain Monte Carlo (MCMC) methods. Despite providing robust 3D reconstructions with limited user supervision (where limited critical parameters are user-defined), these intrinsically iterative methods suffer from a high computational cost (several hours per reconstructed image). Faster alternatives based on convex optimisation tools and spatial regularisation, have been proposed for 3D reconstruction[14–16] but they often require supervised parameter tuning and still need to run several seconds to minutes to converge for a single image. A recent parallel optimization algorithm[17] still reported reconstruction times of the order of seconds. Even the recent algorithm[18] based on a convolutional neural network (CNN) to estimate the scene depth does not meet real-time requirements after training.

Although the single-surface per pixel assumption greatly simplifies the reconstruction problem, it does not hold for complex scenes, for example with cluttered targets, and long-range scenes with larger target footprints. Hence, a second family of methods has been proposed to handle multiple surfaces per pixel[15,19–21]. In this context, 3D reconstruction is significantly more difficult as the number of surfaces per pixel is not a priori known. The earliest methods[21] were based on Bayesian models and so-called reversible-jump MCMC methods (RJ-MCMC) and were mostly designed for single-pixel analysis. Faster optimisation-based methods have also been proposed[15,19], but the recent ManiPoP algorithm[20] combining RJ-MCMC updates with spatial point processes has been shown to provide more accurate results with a

similar computational cost. This improvement is mostly due to ManiPoP's ability to model 2D surfaces in a 3D volume using structured point clouds.

Here we propose a new algorithmic structure, differing significantly from existing approaches, to meet speed, robustness and scalability requirements. As in ManiPoP, the method efficiently models the target surfaces as two-dimensional manifolds embedded in a 3D space. However, instead of designing explicit prior distributions, this is achieved using point cloud denoising tools from the computer graphics community[22]. We extend and adapt the ideas of plug-and-play priors[23–25] and regularisation by denoising[26,27], which have recently appeared in the image processing community, to point cloud restoration. The resulting algorithm can incorporate information about the observation model, e.g., Poisson noise[28], the presence of hot/dead pixels[29,30], or compressive sensing strategies[31,32], while leveraging powerful manifold modelling tools from the computer graphics literature. By choosing a massively parallel denoiser, the proposed method can process dozens of frames per second, while obtaining state-of-the-art reconstructions in the general multiple-surface per pixel setting.

## Results

**Observation model.** A lidar data cube of $N_r \times N_c$ pixels and $T$ histogram bins is denoted by $\mathbf{Z}$, where the photon-count recorded in pixel $(i,j)$ and histogram bin $t$ is $[\mathbf{Z}]_{i,j,t} = z_{i,j,t} \in \mathbb{Z}_+ = \{0, 1, 2, \ldots\}$. We represent a 3D point cloud by a set of $N_\Phi$ points $\Phi = \{(\mathbf{c}_n, r_n) \quad n = 1, \ldots, N_\Phi\}$, where $\mathbf{c}_n \in \mathbb{R}^3$ is the point location in real-world coordinates and $r_n \in \mathbb{R}_+$ is the intensity (unnormalised reflectivity) of the point. A point $\mathbf{c}_n$ is mapped into the lidar data cube according to the function $f(\mathbf{c}_n) = [i, j, t_n]^T$, which takes into account the camera parameters of the lidar system, such as depth resolution and focal length, and other characteristics, such as super-resolution or spatial blurring. For ease of presentation, we also denote the set of lidar depths values by $\mathbf{t} = [t_1, \ldots, t_{N_\Phi}]^T$ and the set of intensity values by $\mathbf{r} = [r_1, \ldots, r_{N_\Phi}]^T$. Under the classical assumption[14,28] that the incoming light flux incident on the TCSPC detector is very low, the observed photon-counts can be accurately modelled by a linear mixture of signal and background photons corrupted by Poisson noise. More precisely, the data likelihood which models how the observations $\mathbf{Z}$ relate to the model parameters can be expressed as

$$z_{i,j,t}|(\mathbf{t}, \mathbf{r}, b_{i,j}) \sim \mathcal{P}\left(\sum_{\mathcal{N}_{i,j}} g_{i,j} r_n h_{i,j}(t - t_n) + g_{i,j} b_{i,j}\right) \quad (1)$$

where $t \in \{1, \ldots, T\}$, $h_{i,j}(\cdot)$ is the known (system-dependent) per-pixel temporal instrumental response, $b_{i,j}$ is the background level in present in pixel $(i,j)$ and $g_{i,j}$ is a scaling factor that represents the gain/sensitivity of the detector. The set of indices $\mathcal{N}_{i,j}$ correspond to the points $(\mathbf{c}_n, r_n)$ that are mapped into pixel $(i,j)$. Figure 1 shows an example of a collected depth histogram.

Assuming mutual independence between the noise realizations in different time bins and pixels, the negative log-likelihood function associated with the observations $z_{i,j,t}$ can be written as

$$g(\mathbf{t}, \mathbf{r}, \mathbf{b}) = -\sum_{i=1}^{N_c} \sum_{j=1}^{N_r} \sum_{t=1}^{T} \log p\left(z_{i,j,t}|\mathbf{t}, \mathbf{r}, b_{i,j}\right) \quad (2)$$

where $p(z_{i,j,t}|\mathbf{t}, \mathbf{r}, b_{i,j})$ is the probability mass associated with the Poisson distribution. This function contains all the information associated with the observation model and its minimisation equates to maximum likelihood estimation (MLE). However,

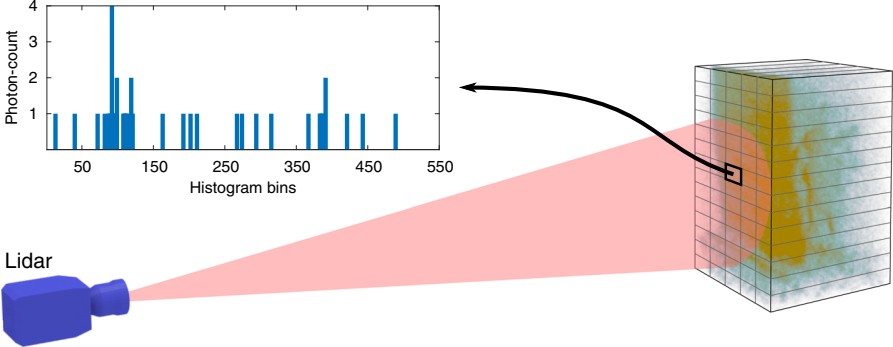

**Fig. 1** Illustration of a single-photon lidar dataset. The dataset consists of a man behind a camouflage net[15]. The graph on the left shows the histogram of a given pixel with two surfaces. The limited number of collected photons and the high background level makes the reconstruction task very challenging. In this case, processing the pixels independently yields poor results, but they can be improved by considering a priori knowledge about the scene's structure

MLE approaches are sensitive to data quality and additional regularisation is required, as discussed below.

**Reconstruction algorithm**. The reconstruction algorithm follows the general structure of PALM[33], computing proximal gradient steps on the blocks of variables **t**, **r** and **b**, as illustrated in Fig. 2. Each update first adjusts the current estimates with a gradient step taken with respect to the log-likelihood (data-fidelity) term $g(\mathbf{t}, \mathbf{r}, \mathbf{b})$, followed by an off-the-shelf denoising step, which plays the role of a proximal operator[34]. While the gradient step takes into account the single-photon lidar observation model (i.e., Poisson statistics, presence of dead pixels, compressive sensing, etc.), the denoising step profits from off-the-shelf point cloud denoisers. A summary of each block update is presented below, whereas an in-detail explanation of the full algorithm can be found in (Supplementary Notes 1–3).

Depth update: A gradient step is taken with respect to the depth variables **t** and the point cloud **Φ** is denoised with the algebraic point set surfaces (APSS) algorithm[35,36] working in the real-world coordinate system. APSS fits a smooth continuous surface to the set of points defined by **t**, using spheres as local primitives (Supplementary Fig. 1). The fitting is controlled by a kernel, whose size adjusts the degree of low-pass filtering of the surface (Supplementary Fig. 2). In contrast to conventional depth image regularisation/denoisers, the point cloud denoiser can handle an arbitrary number of surfaces per pixel, regardless of the pixel format of the lidar system. Moreover, all of the 3D points are processed in parallel, equating to very low execution times.

Intensity update: In this update, the gradient step is taken with respect to **r**, followed by a denoising step using the manifold metrics defined by **Φ** in real-world coordinates. In this way, we only consider correlations between points within the same surface. A low-pass filter is applied using the nearest neighbours of each point (Supplementary Fig. 3), as in ISOMAP[37]. This step also processes all the points in parallel, only accounting for local correlations. After the denoising step, we remove the points with intensity lower than a given threshold, which is set as the minimum admissible reflectivity (normalised intensity) (Supplementary Fig. 4).

Background update: In a similar fashion to the intensity and depth updates, a gradient step is taken with respect to **b**. Here, the proximal operator depends on the characteristics of the lidar system. In bistatic raster-scanning systems, the laser source and single-photon detectors are not co-axial and background counts are not necessarily spatially correlated. Consequently, no spatial regularisation is applied to the background. In this case, the denoising operator reduces to the identity, i.e., no denoising. In monostatic raster-scanning systems and lidar arrays, the

background detections resemble a passive image. In this case, spatial regularisation is useful to improve the estimates (Supplementary Fig. 5). Thus, we replace the proximal operator with an off-the-shelf image denoising algorithm. Specifically, we choose a simple denoiser based on the fast Fourier transform (FFT), which has low computational complexity.

**Large raster-scan scene results**. A life-sized polystyrene head was scanned at a stand-off distance of 40 metres using a raster-scanning lidar system[12]. The data cuboid has size $N_r = N_c = 141$ pixels and $T = 4613$ bins, with a binning resolution of 0.3 mm. A total acquisition time of 1 ms was used for each pixel, yielding a mean of 3 photons per pixel with a signal-to-background ratio of 13. The scene consists mainly of one surface per pixel, with 2 surfaces per pixel around the borders of the head. Figure 3 shows the results for the proposed method, the standard maximum-likelihood estimator and two state-of-the-art algorithms assuming a single[16] or multiple[20] surfaces per pixel. Within a maximum error of 4 cm, the proposed method finds 96.6% of the 3D points, which improves the results of cross-correlation[28], which finds 83.46%, and also performs slightly better than a recent single-surface algorithm[16] and ManiPoP[20], which find 95.2% and 95.23%, respectively. The most significant difference is the processing time of each method: the algorithm only takes 13 ms to process the entire frame, whereas ManiPoP and the single-surface algorithm require 201 s and 37 s, respectively. Whereas a parallel implementation of cross-correlation will almost always be faster than a regularised algorithm (requiring only 1 ms for this lidar frame), the execution time of the proposed method only incurs a small overhead cost while significantly improving the reconstruction quality of single-photon data. The performance of the algorithm was also validated in other raster-scanned scenes (Supplementary Note 7, Supplementary Tables 1 and 2, and Supplementary Figs. 6–8).

**3D Dynamic scenes results**. To demonstrate the real-time processing capabilities of the proposed algorithm, we acquired, using the Kestrel Princeton Lightwave camera, a series of 3D videos (Supplementary Movie 1) with a single-photon array of $N_r = N_c = 32$ pixels and $T = 153$ histogram bins (binning resolution of 3.75 cm), which captures 150,400 binary frames per second. As the pixel resolution of this system is relatively low, we followed a super-resolution scheme, estimating a point cloud of $N_r = N_c = 96$ pixels (Supplementary Fig. 9). This can be easily achieved by defining an undersampling operation in $f(\cdot)$, which maps a window of $3 \times 3$ points in the finest resolution (real-world coordinates) to a single pixel in the coarsest resolution (lidar coordinates). As processing a single lidar frame with the method takes

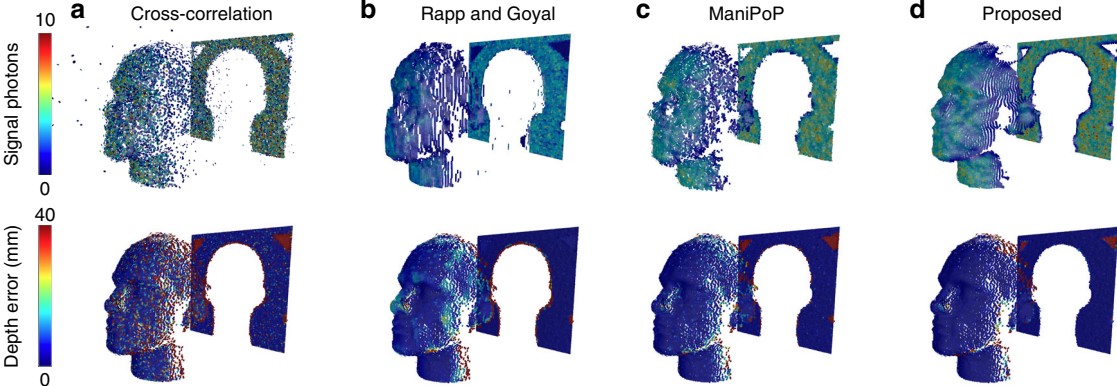

**Fig. 2** Block diagram of the proposed real-time framework. The algorithm iterates between depth, intensity and background updates, applying a gradient step followed by a denoiser. Each step can be processed very quickly in parallel, resulting in a low total execution time

**Fig. 3** Comparison of 3D reconstruction methods. Reconstruction results of **a** cross-correlation, **b** Rapp and Goyal[16], **c** ManiPoP[20] and **d** the proposed method. The colour bar scale depicts the number of returned photons from the target assigned to each 3D point. Cross-correlation does not include any regularisation, yielding noisy estimates, whereas the results of Rapp and Goyal, ManiPoP and the proposed method show structured point clouds. The method of Rapp and Goyal correlates the borders of the polystyrene head and the backplane (as it assumes a single surface per pixel), whereas ManiPoP and the proposed method do not promote correlations between them

20 ms, we integrated the binary acquisitions into 50 lidar frames per second (i.e., real-time acquisition and reconstruction). At this frame rate, each lidar frame is composed of 3008 binary frames.

Figure 4 shows the imaging scenario, which consists of two people walking between a camouflage net and a backplane at a distance of ~320 metres from the lidar system. Each frame has ~900 photons per pixel, where 450 photons are due to target returns and the rest are related to dark counts or ambient illumination from solar background. Most pixels present two surfaces, except for those in the left and right borders of the camouflage, where there is only one return per pixel. A maximum number of three surfaces per pixel can be found in some parts of the contour of the human targets.

## Discussion

We have proposed a real-time 3D reconstruction algorithm that is able to obtain reliable estimates of distributed scenes using very few photons and/or in the presence of spurious detections. The proposed method does not make any strong assumptions about the 3D surfaces to be reconstructed, allowing an unknown number of surfaces to be present in each pixel. We have demonstrated similar or better reconstruction quality than other existing methods, while improving the execution speed by a factor up to $10^5$. We have also demonstrated the reliable real-time 3D reconstruction of scenes with multiple surfaces per pixel at long distance (320 m) and high frame rates (50 frames per second) in daylight conditions. The method can be easily implemented for general purpose graphical processing units (GPGPU)[38], and thus is compatible with use in modern embedded systems (e.g., self-driving cars). Minimal operating conditions (i.e., minimum signal-to-background ratio and photons per pixel required to ensure good reconstruction with high probability) are discussed

in (Supplementary Note 5 and Supplementary Fig. 10). The algorithm combines a priori information on the observation model (sensor statistics, dead pixels, sensitivity of the detectors, etc.) with powerful point cloud denoisers from computer graphics literature, outperforming methods based solely on computer graphics or image processing techniques. Moreover, we have shown that the observation model can be easily modified to perform super-resolution. It is worth noting that the proposed model could also be applied to other scenarios, e.g., involving spatial deblurring due to highly scattering media. While we have chosen the APSS denoiser, the generality of our formulation allows us to use many point cloud (depth and intensity) and image (background) denoisers as building blocks to construct other variants. In this way, we can control the trade-off between reconstruction quality and computing speed (Supplementary Note 6). Finally, we observe that the proposed framework can also be easily extended to other 3D reconstruction settings, such as sonar[39] and multispectral lidar[32].

## Methods

**3D Reconstruction algorithm**. The reconstruction algorithm has been implemented on a graphics processing unit (GPU) to exploit the parallel structure of the update rules. Both the initialisation (Supplementary Note 3, Supplementary Figs. 11 and 12) and gradient steps process each pixel independently in parallel, whereas the point cloud and intensity denoising steps process each world-coordinates pixel in parallel, making use of the GPU shared memory to gather information of neighbouring points (Supplementary Note 4). The background denoising step is performed using the CuFFT library[38]. The algorithm was implemented using the parallel programming language CUDA C++ and all the experiments were performed using an NVIDIA Xp GPU. The surface fitting was performed using the Patate library[40].

Figure 5 shows the execution time per frame as a function of the total number of pixels and the mean active bins per pixel (i.e., the number of bins that have one or more photons) for the mannequin head dataset of Fig. 3. For image sizes smaller than 150 × 150, the algorithm has approximately constant execution time, due to

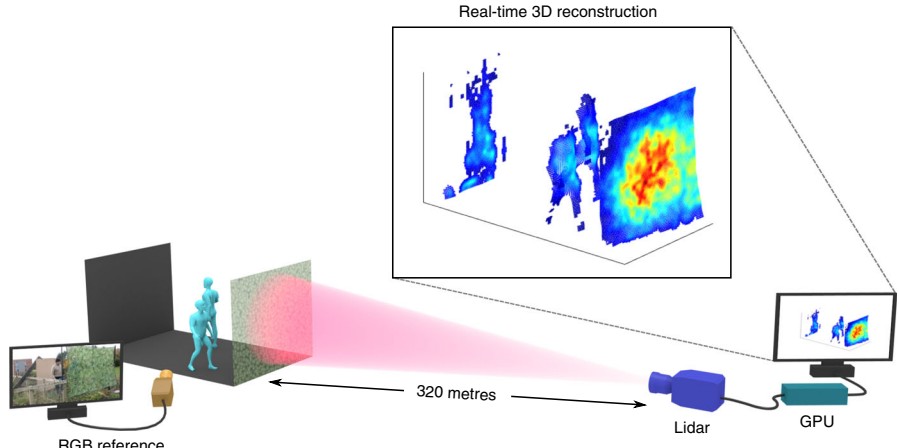

Real-time 3D reconstruction

RGB reference

320 metres

Lidar

GPU

**Fig. 4** Schematic of the 3D imaging experiment. The scene consists of two people walking behind a camouflage net at a stand-off distance of 320 metres from the lidar system. An RGB camera was positioned a few metres from the 3D scene and used to acquire a reference video. The proposed algorithm is able to provide real-time 3D reconstructions using a GPU. As the lidar presents only $N_r = N_c = 32$ pixels, the point cloud was estimated in a higher resolution of $N_r = N_c = 96$ pixels (Supplementary Movie 1)

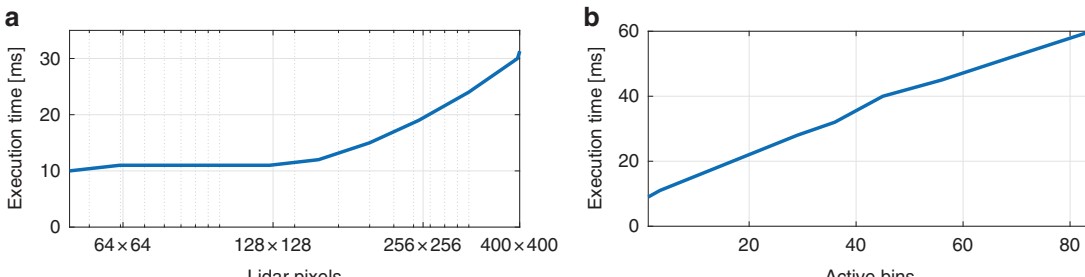

**Fig. 5** Execution time of the proposed method. The execution time is shown as function of **a** lidar pixels (having a mean of 4 active bins per pixel), and **b** histogram bins with non-zero counts, for an array of $141 \times 141$ pixels. As all the steps involved in the reconstruction algorithm can process the pixel information in parallel, the total execution time does not increase significantly when more pixels are considered. However, as the pixel-wise operations are not fully parallel, there is a linear dependency on the number of active (non-zero) bins present in the lidar frame

the completely parallel processing of pixels. Larger images yield an increased execution time, as a single GPU does not have enough processors to handle all pixels at the same time (and other memory read/write constraints). As the per-pixel computations are not parallelised, the algorithm shows an approximately linear dependence with the mean number of active bins per pixel (Supplementary Note 4).

**Imaging set-up (dynamic scenes)**. Our system used a pulsed fibre laser (by BKtel, HFL-240am series) as the source for the flood illumination of the scene of interest. This had a central wavelength of 1550 nm and a spectral full width half maximum (FWHM) of ~9 nm. The output fibre from the laser module was connected to a reflective collimation package and the exiting beam then passed through a beam expander arrangement consisting of a pair of lenses. The lenses were housed in a zoom mechanism that enabled the diameter of the illuminating beam at the scene of interest to be adjusted to match the field of view of the camera (Supplementary Methods, Supplementary Fig. 13).

We used a camera with a $32 \times 32$ array of pixels for the depth and intensity measurements reported here. This camera (by Princeton Lightwave Incorporated, Kestrel model) had an InGaAs/InP SPAD detector array with the elements on a 100 $\mu$m square pitch, resulting in an array with active area dimensions of ~3.2 × 3.2 mm. At the operating wavelength of 1550 nm, the elements in the array had a quoted photon detection efficiency of ~25% and a maximum mean dark count rate of ~320 kcps. The camera was configured to operate with 250 ps timing bins, a gate duration of 40 ns, and a frame rate of 150 kHz (this was close to the expected maximum frame rate of the camera). The camera provided this 150 kHz electrical clock signal for the laser, and the average optical output power from the laser at this repetition rate was ~220 mW and the pulse duration was ~400 ps. The camera recorded data continuously to provide a stream of binary frames at a rate of 150,400 binary frames per second.

An f/7, 500 mm effective focal length lens (designed for use in the 900–1700 nm wavelength region) was attached to the camera to collect the scattered return photons from the scene. This resulted in a field of view of ~0.5 arc degrees. As these

measurements were carried out in broad daylight, a set of high performance passive spectral filters was mounted between the rear element of the lens and the sensor of the camera in order to minimise the amount of background light detected.

Our optical setup was a bistatic arrangement—the illuminating transmit channel and the collecting receive channel had separate apertures, i.e., the two channels were not co-axial. This configuration was used in order to avoid potential issues that could arise in a co-axial (monostatic) system due to back reflections from the optical components causing damage to the sensitive focal plane array. The parallax inherent in the bistatic optical configuration meant that a slight re-alignment of the illumination channel, relative to the receive (camera) channel, was required for scenes at different distances from the system.

## Data availability

The lidar data used in this paper are available in the repository https://gitlab.com/Tachella/real-time-single-photon-lidar.

## Code availability

A cross-platform executable file containing the real-time method is available in the repository https://gitlab.com/Tachella/real-time-single-photon-lidar. The software requires an NVIDIA GPU with compute capability 5.0 or higher.

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

## Acknowledgements

This work was supported by the Royal Academy of Engineering under the Research Fellowship scheme RF201617/16/31 and by the Engineering and Physical Sciences Research Council (EPSRC) Grants EP/N003446/1, EP/M01326X/1, EP/K015338/1, and EP/S000631/1, and the MOD University Defence Research Collaboration (UDRC) in Signal Processing. We gratefully acknowledge the support of NVIDIA Corporation with the donation of the Titan Xp GPU used for this research. We also thank Bradley Schilling of the US Army RDECOM CERDEC NVESD and his team for their assistance with the field trial (camouflage netting data). Finally, we thank Robert J. Collins for his help with the measurements made of the scene at 320 metres, and David Vanderhaeghe and Mathias Paulin for their help on volumetric rendering.

## Author contributions

J.T. performed the data analysis, developed and implemented the computational reconstruction algorithm; Y.A. and N.M. contributed to the development of the reconstruction algorithm; A.M., R.T. and G.S.B. developed the experimental set-up; A.M. and R.T. performed the data acquisition and developed the long-range experimental scenarios; Y.A., G.S.B., J.-Y.T. and S.M. supervised and planned the project. All authors contributed to writing the paper.

## Competing interests

The authors declare no competing interests.
