## [Peer Review File · Nature Communications]

Peer Review

Reviewer #1 (Remarks to the Author):

Summary of key results:

The submitted manuscript introduces a computational approach to reconstructing 3D scenes from single-photon lidar data. This approach combines detailed physics-based probabilistic modeling of measurement processes [the "image processing" of the title, though it is arguable whether many readers would make this association] with techniques for point-cloud denoising [the "computer graphics" of the title]. The point-cloud denoising techniques are more effective than the sorts of regularization techniques that have been commonly applied and also much more easily scalable to large problem sizes than the Markov chain Monte Carlo (MCMC) approach (from the same group) that represents the state of the art for achieving high reconstruction quality (accuracy) with relatively little regard for computational complexity.

As acknowledged in the manuscript, the method developed here is philosophically aligned with the "plug-and-play" and "regularization by denoising" (RED) approaches that have emerged recently for solving inverse problems in imaging. The authors' application of this philosophy to 3D imaging is new and creative. In particular, one might imagine the use of off-the-shelf image denoisers applied to depth images as a route to developing a plug-and-play or RED approach to single-photon lidar. What is done here is more novel than that because there is a larger leap between point-cloud denoising and interpretation as a prior. In any case, the techniques are strikingly successful and well-described and well-illustrated.

In my opinion, the most negative that one could be about this work is to call the quality improvements over Rapp and Goyal (for the single-object per pixel case, Fig. 3, Supplementary Fig. 4) and over ManiPoP (for single- or multiple-objects per pixel, Fig. 3, Supplementary Figs. 4-6) modest. However, I believe that the quality improvement combined with the enormous computational speedup and conceptual novelty make this manuscript important to researchers and practitioners in the field. It could stand as a benchmark for many years. If it does not, this will likely be because it inspired important subsequent work. I enthusiastically recommend publication in Nature Communications.

Criticisms of omission or lack of emphasis:

1. A well-designed computational imaging method, such as described in the submitted manuscript, depends on the combination of algorithm, hardware, and implementation. Still, I would like to be able to understand how much of the speedup is due to the parallelizable approach being well-implemented on suitable hardware. Is there some way to describe this? Would there any speedup with an ordinary serial architectures? The conventional wisdom is that MCMC methods have high complexity, so improvement over ManiPoP would presumably persist.

2. The experimental work is strong, especially the breadth and clarity of the comparisons with previous methods. Still, there is room for improvement and expansion. I do not consider these to be necessary for publication, but the authors may find that they can address some of these points without extraordinary delay.

First, it would be nice for the authors to discuss sensitivity to certain parameters (or evidence of lack of sensitivity) such as the threshold in "we remove the points with intensity lower than a given threshold". That threshold presumably affects both the reconstruction quality and the computation time. There also are regularization parameters.

Second, it would be nice for the authors to discuss trends (e.g., whether the comparisons become

more or less favorable) with changes in the environmental conditions. Lower signal strength could be emulated by using subsets of data. Lower signal-to-background ratio could be emulated by introducing additional pseudorandom detection times.

Stylistic suggestions and minor corrections:

ManiPoP: Your SIAM J. Imaging Sci. paper uses mixed case for ManiPoP. Here you are inconsistent.

Title: I am signing this review, and my own most cited paper has a subtitle with "meets" in it. Still, I think your use of "meets" is not clear. To many people, a distinction between computational imaging and image processing is that image processing uses Gaussian likelihoods (or no discussion of likelihoods at all) and priors that are generically appropriate for natural images. Thus, your use of "image processing" would not evoke the detailed probabilistic modeling of single-photon lidar data. I think "computational imaging" or "signal processing" would be more appropriate than "image processing".

Abstract: I find the phrase "reconstruction of single-photon data" to be slightly awkward because "single-photon data" is not what is reconstructed. The sentence with "combining statistical models" is ambiguously phrased. If you are asserting that those models are "from the computer graphics community", it seems an incorrect sentence; else, it seems dangling that you are not explicitly attributing the models to the other counterpart in your title ("image processing").

Body paragraph 1: "relative slow scanned" -> "relatively slowly scanned" and delete "after the event".

Body paragraph 2: I find "single surface per lidar waveform" to be less clear than something like "single surface per transverse pixel". Also, I suggest a comma after "differing significantly from existing approaches".

Following "Observation model" heading: It seems to me a typo to write that $z_{\{i,j,t\}}$ must be positive rather than merely nonnegative. In that same paragraph, you may decide to use only one out of "intensity" and "reflectivity" here and throughout the paper.

Figure 3: You may want to consider whether there is a logic to the order of columns. I would expect cross-correlation to be first as the most basic. (You could similarly reconsider for some of the figures in the Supplementary material.)

Methods paragraph 1: The first sentence is awkward because "can be run completely" seems redundant. Also, correct "paralleling".

Methods paragraph 4: The dark count of rate of 50 kHz does not match the 320 kcps given in the Supplementary material. Have I misunderstood some change of configuration between the two?

Data availability: "free" -> "freely"

Supplementary sentence containing (10): First, (10) is missing something. Second, I think it is inconsistent with your own notation to write $(i,j,t_n)^T$ because I think a tuple written with parentheses is not to be interpreted as a row or a column. (My point is that I would expect you to write this as $[i,j,t_n]^T$.) Finally, it is frustrating to have no explanation of the choice of $w(t)$.

Setting the stepsizes paragraph 1: Doubled "the". Also, missing a comma before "are negligible".

Supplementary sentence containing (15): The equation looks a little strange because σ^2 could be factored out.

Bullet before Algorithm 1: The first sentence is flawed.

Last sentence before Additional results: "less" -> "fewer"

Below Algorithm 3: "time is presented" -> "times are presented"

-Vivek Goyal

Reviewer #2 (Remarks to the Author):

Summary

The authors present an approach for single-photon LIDAR which reconstructs 3D geometry with multiple depths per spatial location at high framerates. Real-time data acquisition is demonstrated using a flash LIDAR system with a 2D SPAD array, and use of efficient denoising priors in a GPU implementation enables fast reconstruction.

Strengths

- + The paper is mostly very clear and well-written. The results and figures are explained well, and the additional experimental results and explanations provided in the supplemental material are appreciated.
- + The results appear to show performance comparable to state-of-the-art single-photon LIDAR reconstruction algorithms (which can also reconstruct multiple surfaces per spatial location, e.g. MANIPOP), but at a significantly reduced computational cost.
- + The main contribution of real-time reconstruction rates with the ability to recover multiple surfaces per spatial location would be appreciated by the community and could enable new applications for single-photon LIDAR.

Comments

The idea of using efficient denoising priors in an alternating update optimization framework for single-photon LIDAR was also demonstrated by Heide et al. (2018) in "Sub-picosecond photon-efficient 3D imaging using single-photon sensors". Heide et al. note that a GPU implementation of their framework should achieve roughly similar reconstruction times (~ 5 ms) as what the authors demonstrate in their work. While one difference is that Heide et al. focus on depth estimation in the pileup regime with minimal background counts, the authors should cite and include a discussion of this work as well, especially given the similarities in the image formation models and reconstruction procedures.

The authors use the APSS denoiser for the depth update step. I would recommend additional discussion about the motivation and tradeoffs for this choice, as well as potential alternatives.

The authors mention that background counts recorded in a bistatic lidar system are not necessarily spatially correlated, but in a mono-static system, the background detections resemble a passive image (with spatial correlations). However, my understanding is that in the authors' own bistatic system, the SPAD array would still capture a passive image from the background counts. Please clarify.

With respect to the timings reported for the reconstruction algorithm, I was confused why the $141 \times 141 \times 4613$ data cuboid used for the raster-scan result took only 13 ms to process while the significantly smaller $32 \times 32 \times 153$ resolution volume takes 20 ms to process. The former input volume is nearly 600 times larger, so it seemed unusual that the reconstruction would run almost twice as fast as for the smaller volume. Please clarify.

The authors report that the 2D SPAD array captures 150,400 binary frames per second. Does this mean that the sensor provides a maximum output rate of 150k histograms per second? Is there a limit of how many photon timestamps can be captured per histogram?

I wasn't entirely convinced about the benefit of the upsampling from 32x32 spatial resolution to 96x96 demonstrated in the dynamic results. Do the super-resolved results show much improvement over a naive upscaling of a native 32x32 reconstruction?

For Fig. 5 of the methods section, please clarify how many active bins are used in the execution time vs LIDAR pixel comparison and how many LIDAR pixels are used in the execution time vs active bins comparison.

Analysis

The numerical analyses presented in the paper seem appropriate and valid.

Reproducibility

The level of detail provided seems adequate to reproduce the work.

1 Response to reviewer #1

Comment 1.1

As acknowledged in the manuscript, the method developed here is philosophically aligned with the "plug-and-play" and "regularization by denoising" (RED) approaches that have emerged recently for solving inverse problems in imaging. The authors' application of this philosophy to 3D imaging is new and creative. In particular, one might imagine the use of off-the-shelf image denoisers applied to depth images as a route to developing a plug-and-play or RED approach to single-photon lidar. What is done here is more novel than that because there is a larger leap between point-cloud denoising and interpretation as a prior. In any case, the techniques are strikingly successful and well-described and well-illustrated.

Response:

We would like to thank the reviewer for his comment.

Comment 1.2

In my opinion, the most negative that one could be about this work is to call the quality improvements over Rapp and Goyal (for the single-object per pixel case, Fig. 3, Supplementary Fig. 4) and over ManiPoP (for single- or multiple-objects per pixel, Fig. 3, Supplementary Figs. 4-6) modest.

Response:

In this work, we have focused on exploiting parallel denoising algorithms such as APSS to achieve the real-time benchmark, which we believe to be the limiting factor of previous 3D reconstruction algorithms. However, the core idea of this paper of using point cloud denoisers as 3D regularisers can be extended to other off-the-shelf denoisers to further improve the quality of this method. These alternatives are discussed in comment 2.2. A discussion has been included in the section ‘Beyond APSS: point cloud denoising alternatives’ of the updated supplementary material.

Comment 1.3

1. A well-designed computational imaging method, such as described in the submitted manuscript, depends on the combination of algorithm, hardware, and implementation. Still, I would like to be able to understand how much of the speedup is due to the parallelizable approach being well-implemented on suitable hardware. Is there some way to describe this? Would there any speedup with an ordinary serial architectures? The conventional wisdom is that MCMC methods have high complexity, so improvement over ManiPoP would presumably persist.

Response:

The proposed method relies on the combination of 1) the parallel update structure, which is well-tuned for GPUs or similar single instruction multiple data (SIMD) architectures, 2) the fast convergence of the PALM strategy [6], which requires few iterations to achieve a local minima and 3) the reduced number of parameters to be estimated.

Clearly, the most important speed-up is provided by the parallel updates. However, as observed by the reviewer, a serial implementation of the proposed method would be faster than MCMC-based methods such as ManiPoP [7], as the total number of computations is smaller. The proposed method runs for 25 PALM iterations, whereas ManiPoP is intrinsically sequential (the RJ-MCMC structure hinders parallel updates) and requires more iterations to explore the full posterior distribution. However, the speed-up is also due to the PALM-structure and reduced number of unknown parameters compared to existing multi-depth algorithms based on convex relaxation techniques, such as [8] or [9]. These methods require the storage of an intensity cube with the same dimensions as the entire lidar data of order $\mathcal{O}(N_r N_c T)$, whereas the proposed method only stores in memory the point positions, intensities and background levels of order $\mathcal{O}(N_r N_c)$, which are significantly smaller than the full Lidar cube. We have included a discussion of memory requirements in the implementation section of the updated supplementary material.

Comment 1.4

2. The experimental work is strong, especially the breadth and clarity of the comparisons with previous methods. Still, there is room for improvement and expansion. I do not consider these to be necessary for publication, but the authors may find that they can address some of these points without extraordinary delay.

Response:

We have improved the experimental work of this paper, studying the selection of the hyperparameters (see comment 1.5) and evaluating the performance as a function of the signal-to-background ratio and mean number photons per pixel (see comment 1.6). We believe that these additional experiments have improved the quality of the paper, providing guidelines to select hyperparameter values and quantifying the operation limits of the algorithm.

Comment 1.5

First, it would be nice for the authors to discuss sensitivity to certain parameters (or evidence of lack of sensitivity) such as the threshold in "we remove the points with intensity lower than a given threshold". That threshold presumably affects both the reconstruction quality and the computation time.

Response:

Following the reviewer comment, we have performed additional experiments regarding the choice of hyperparameters using the "polystyrene head without backplane" dataset (details of this dataset are found in Table 2 of the supplementary material). The figure below shows the number of true and false detections as a function of the intensity threshold. As we increase the threshold, the number of true detections decreases monotonically. In contrast, the number of false detections increases exponentially as the threshold tends to zero. The best performing values are between 0.2 and 0.4 photons, coinciding with the reflectivity interval from 5% to 10%. This interval can be used as a guideline for setting r_{\min} . The execution time is not affected significantly by the threshold, as the complexity is mostly driven by the (fixed) number of photons.

Reconstruction performance as a function of the intensity threshold for the polystyrene head with backplane. The best performing values have a normalized intensity between 5% and 10%.

The reflectivity update depends on the amount of filtering β , which mostly impacts the intensity estimation. The figure below shows the intensity absolute error (as defined in [10]) as a function of $\beta \in [0, 1]$. Very small values of β mean negligible filtering, finding less points and resulting in a larger intensity error. Large values of β (close to 1) oversmooth the estimates, generating false detections and also resulting in a larger intensity error (this effect is reduced by the very smooth profile of a polystyrene head). Good values for β generally lie in the interval [0.1, 0.3].

Effect of the amount of low-pass filtering on the reconstruction quality. Large values oversmooth the estimates, generating false detections and also incurring in a larger intensity error, whereas low values do not impose sufficient spatial correlation, reducing the number of true detections.

The depth update depends on d_t , the APSS kernel size in the depth direction. The figure below shows the impact of d_t in terms of true and false detections and mean depth absolute error (DAE). Small values of d_t result in poor reconstructions, as the kernel is too small to correlate neighbouring points, whereas large values oversmooth the depth estimates and may also mix different surfaces. The best choice lies around 8 and 10, which also has a physically interpretation, as discussed in comment 1.6.

Effect of the APSS kernel size in the depth direction on the reconstruction quality. Low values fail to correlate neighbouring points, whereas large values oversmooth the depth estimates.

The background update depends on the hyperparameter λ_L , which controls the degree of correlation between neighbouring background levels. The figure below shows the background estimation performance (measured by the NMSE) as a function of λ_L for the “polystyrene head without backplane”. While low values of λ_L do not impose sufficient correlation, large values of λ_L tend to oversmooth the estimates. Although the best choices lie in the interval $[0.5, 2]$, the performance is not very sensitive to bad specifications of λ_L .

Effect of the amount of background regularisation λ_L on the estimation of background levels.

These experiments have been included in the section ‘Improvements by upsampling in small lidar arrays’ of the updated supplementary material and has been mentioned in the revised paper.

Comment 1.6

Second, it would be nice for the authors to discuss trends (e.g., whether the comparisons become more or less favorable) with changes in the environmental conditions. Lower signal strength could be emulated by using subsets of data. Lower signal-to-background ratio could be emulated by introducing additional pseudorandom detection times.

Response:

We generated 100 synthetic lidar cubes for mean SBR values in $[0.01, 100]$ and mean photons per pixels in $[0.1, 100]$, using the ground truth point cloud, data cube size and impulse response from the polystyrene head without backplane dataset. As a baseline, we compared the with proposed method the standard cross-correlation algorithm. To account for the pixels without objects, we post-processed the output of cross-correlation by removing points below a normalized intensity of 10%. We considered the number of true and false detections, depth absolute error (only computed for true detections and reconstructions with more than 80% of detected points), intensity absolute error (normalised by the mean signal photon counts to approximately lie between 0 and 1) and background NMSE. The results obtained are gathered in the figure below. The proposed method performs well in a wider range of conditions, achieving reconstructions with ≈ 0.1 photons per pixel and up to signal-to-noise background ratio of 0.01 (with 100 photons per pixel or more). Moreover, cross-correlation generates many orders of magnitude more false detections than the new method. Interestingly, the proposed algorithm exhibits a sharper transition in the detection of true points, meaning that, for a given signal-to-background ratio, either none or most of the points will be found depending on the recorded photon count. The new method achieves smaller depth and intensity absolute errors than cross-correlation in all conditions, as it exploits the manifold structure of the scene. The proposed method also achieves a significantly smaller background NMSE, capturing the spatial correlation in the background image.

Comparison of the proposed method and cross-correlation with thresholding in a target detection setting for different SBR and mean photons per pixel values. The depth absolute error is only displayed for reconstructions with more than 80% and is left blank otherwise.

This experiment has been included in the section ‘Operation boundary conditions’ of the updated supplementary material and has been mentioned in the revised paper.

Comment 1.7

ManiPoP: Your SIAM J. Imaging Sci. paper uses mixed case for ManiPoP. Here you are inconsistent.

Response:

We have corrected the reference to ManiPoP throughout the paper, only using the mixed case ManiPoP.

Comment 1.8

Title: I am signing this review, and my own most cited paper has a subtitle with "meets" in it. Still, I think your use of "meets" is not clear. To many people, a distinction between computational imaging and image processing is that image processing uses Gaussian likelihoods (or no discussion of likelihoods at all) and priors that are generically appropriate for natural images. Thus, your use of "image processing" would not evoke the detailed probabilistic modeling of single-photon lidar data. I think "computational imaging" or "signal processing" would be more appropriate than "image processing".

Response:

We agree with the reviewer on this comment. We have replaced the term 'image processing' with 'computational imaging', to stress the fact that computer graphics denoising algorithms are used here as part of the image formation process, applying them in the context of inverse problems. The new title reads 'Real-Time 3D Reconstruction of Complex Scenes Using Single-Photon lidar: When Computational Imaging Meets Computer Graphics'.

Comment 1.9

Abstract: I find the phrase "reconstruction of single-photon data" to be slightly awkward because "single-photon data" is not what is reconstructed. The sentence with "combining statistical models" is ambiguously phrased. If you are asserting that those models are "from the computer graphics community", it seems an incorrect sentence; else, it seems dangling that you are not explicitly attributing the models to the other counterpart in your title ("image processing").

Response:

We have updated the abstract to better highlight the contributions of this paper (real-time performance of 20 ms using standard GPUs, the general multiple-surface per pixel formulation and the experimental validation at broad daylight from distances up to 320 metres). Also, we have modified "combining statistical models and highly scalable computational tools from the computer graphics community", to "combining statistical models with highly scalable computational tools from the computer graphics community".

Comment 1.10

Body paragraph 1: "relative slow scanned" -> "relatively slowly scanned" and delete "after the event". Body paragraph 2: I find "single surface per lidar waveform" to be less clear than something like "single surface per transverse pixel". Also, I suggest a comma after "differing significantly from existing approaches".

Response:

We have applied these corrections to the updated manuscript.

Comment 1.11

Following "Observation model" heading: It seems to me a typo to write that $z_{i,j,t}$ must be positive rather than merely nonnegative. In that same paragraph, you may decide to use only one out of "intensity" and "reflectivity" here and throughout the paper.

Response:

Indeed, $z_{i,j,t}$ can equal zero, and the typo in the definition of \mathbb{Z}_+ has been corrected. We have also replaced the word 'reflectivity' with 'intensity' (meaning unnormalized reflectivity) throughout the paper for consistency.

Comment 1.12

Figure 3: You may want to consider whether there is a logic to the order of columns. I would expect cross-correlation to be first as the most basic. (You could similarly reconsider for some of the figures in the Supplementary material.)

Response:

We have modified the order of the columns according to the year of publication of the evaluated methods, throughout the main paper and supplementary material.

Comment 1.13

Methods paragraph 1: The first sentence is awkward because "can be run completely" seems redundant. Also, correct "paralleling".

Response:

We thank the reviewer for spotting these typos, which have been corrected in the updated manuscript.

Comment 1.14

Methods paragraph 4: The dark count of rate of 50 kHz does not match the 320 kcps given in the Supplementary material. Have I misunderstood some change of configuration between the two?

Response:

There was indeed a mistake in the methods paragraph, the dark count rate should be 320 kcps (as in the Supplementary Material), which has been corrected in the updated manuscript.

Comment 1.15

Data availability: "free" -> "freely". Setting the stepsizes paragraph 1: Doubled "the". Also, missing a comma before "are negligible". Last sentence before Additional results: "less" -> "fewer" Below Algorithm 3: "time is presented" -> "times are presented" Bullet before Algorithm 1: The first sentence is flawed.

Response:

These corrections have been included in the new manuscript. The sentence in the bullet before Algorithm 1 has been corrected to "Histograms collected using single-photon Lidar systems with high temporal resolution ($< 20\text{ps}$), e.g., raster-scanning systems, generally present a large number of sparsely populated bins, hindering any dense computations using the Anscombe transform."

Comment 1.16

Supplementary sentence containing (10): First, (10) is missing something. Second, I think it is inconsistent with your own notation to write $(i, j, t_n)^T$ because I think a tuple written with parentheses is not to be interpreted as a row or a column. (My point is that I would expect you to write this as $[i, j, t_n]^T$.) Finally, it is frustrating to have no explanation of the choice of $w(t)$.

Response:

Equation (10) had a mistake in the previous version, which has been corrected to

$$\arg \min_{\mathbf{u}} \sum_{n=1}^{N_{\Phi}} w(\|\mathbf{c}_n - \mathbf{c}_r\|_{\Sigma}) \phi_{\mathbf{u}}^2(\mathbf{c}_r). \quad (1)$$

As suggested by the reviewer, we have corrected the notation $(i, j, t_n)^T$ to $[i, j, t_n]^T$ throughout the paper, where the quantity is interpreted as a (column) vector. The kernel $w(t)$ was chosen with diagonal entries, i.e.,

$$\Sigma = \begin{pmatrix} d_x & 0 & 0 \\ 0 & d_y & 0 \\ 0 & 0 & d_t \end{pmatrix} \quad (2)$$

. In all the experiments, we set $d_x = d_y = 1$, such that only the 8 closest neighbouring pixels have strong weights, and d_t to be the minimum distance between two surfaces in the same transverse pixel, which is chosen according to the bin width of the lidar system to have a physical meaning. Interestingly, we chose the same distance as the hard constraint between points in the same pixel in ManiPoP [7]. This explanation has been included in the updated manuscript (supplementary material).

Comment 1.17

Supplementary sentence containing (15): The equation looks a little strange because σ^2 could be factored out.

Response:

We have factored out σ^2 from (15). The updated equation is

$$L_t^s \leq \frac{1}{\sigma^2} \max_{i,j} \sum_{t=1}^T z_{i,j,t}. \quad (3)$$

2 Response to reviewer #2

Comment 2.1

The idea of using efficient denoising priors in an alternating update optimization framework for single-photon LIDAR was also demonstrated by Heide et al. (2018) in "Sub-picosecond photon-efficient 3D imaging using single-photon sensors". Heide et al. note that a GPU implementation of their framework should achieve roughly similar reconstruction times (5 ms) as what the authors demonstrate in their work. While one difference is that Heide et al. focus on depth estimation in the pileup regime with minimal background counts, the authors should cite and include a discussion of this work as well, especially given the similarities in the image formation models and reconstruction procedures.

Response:

We thank the reviewer for pointing out this reference. We have included this work in the introduction of the paper. However, our formulation is conceptually different than the work in [1], as we propose a novel approach that can process general 3D surfaces, whereas the model [1] estimates depth and reflectivity images, being applicable only in scenes with exactly one surface per pixel. As pointed out by reviewer #1 in comment 1.1, the method presented here introduces a novel interpretation of point cloud denoisers as 3D priors, dropping the constraint and approximation of depth images (and related depth image denoisers). The novel idea of this work departs from these assumptions, suggesting a new family of methods based on point cloud denoising algorithms.

The total variation regulariser used in [1] for the depth image has been explored in multiple works in single-photon lidar [2-5]. For example, the optimization algorithm introduced in [4] shares the same regularisers and should reach a similar solution in the low-flux regime if the impulse response of the system is Gaussian. We expect these methods to perform worse in presence of background illumination than the unmixing algorithm [5], as they do not estimate high background levels (although the main paper in [1] claims they do, the supplementary material and code provided assume a fixed background). We consider that a comparison with [5] is more relevant as a reference of a single-depth algorithm.

Regarding the execution time, the work in [1] reports an average execution time of 100 seconds using a GPU and conjectures an execution time of 5 ms although is not demonstrated experimentally in [1]. Our parallel implementation of cross-correlation required greater than 50 ms to process the dense datasets of size $150 \times 150 \times 1101$ for the initialization of [1]. We downloaded the publicly available code of [1], and could only obtain execution times of 220 seconds or greater for the datasets considered in that paper. The computer used to ran the code was the same that obtained the execution times in this paper (Intel i7 CPU, 16 GB RAM memory, Titan Xp GPU, MATLAB 2018a).

Comment 2.2

The authors use the APSS denoiser for the depth update step. I would recommend additional discussion about the motivation and tradeoffs for this choice, as well as potential alternatives.

Response:

In this work, we focus on the APSS denoiser to target real-time performance, profiting from the parallel structure and closed-form updates. However, we could imagine other choices with different trade-offs between execution time, memory requirement and reconstruction quality [11]. For example, a straightforward alternative is the simple point set surface (SPSS) denoiser instead of APSS. The proposed method provides a framework to incorporate different types of prior information, avoiding the need to develop specific algorithms for single-photon lidar. As explained in [12], APSS only relies on a local surface smoothness prior, whereas more sophisticated denoisers exploit more complex prior knowledge on the point cloud structure. If we want to capture non-local correlations between point cloud patches, we could use the denoiser in [13], which uses a dictionary learning approach. Higher-level knowledge on the scene, such as the presence of buildings or humans could be also exploited through dedicated denoisers. The algorithm of [14] uses planes to denoise point clouds of building facades, being adapted for remote sensing/outdoor applications. Finally, we could also profit from available 3D data using data-driven denoisers. In this direction, we can use algorithms that fit templates of possible objects [15] or profit from recent advances in graph convolutional neural networks [16], which are specially designed to handle point cloud structures [17, 18].

This discussion has been included in the section ‘Beyond APSS: point cloud denoising alternatives’ of the updated supplementary material.

Comment 2.3

The authors mention that background counts recorded in a bistatic lidar system are not necessarily spatially correlated, but in a mono-static system, the background detections resemble a passive image (with spatial correlations). However, my understanding is that in the authors’ own bistatic system, the SPAD array would still capture a passive image from the background counts. Please clarify.

Response:

The Princeton Lightwave 32×32 used in the 3D video experiments had a bi-static configuration, however we agree with the reviewer that the background counts resemble a passive image. In contrast, bi-static raster-scanning systems (e.g., [8]), have background counts that do not resemble a passive image, as the detector’s footprint covers the whole imaged scene. This explanation has been included in the ‘Background update’ section of the updated paper and supplementary material.

Comment 2.4

With respect to the timings reported for the reconstruction algorithm, I was confused why the $141 \times 141 \times 4613$ data cuboid used for the raster-scan result took only 13 ms to process while the significantly smaller $32 \times 32 \times 153$ resolution volume takes 20 ms to process. The former input volume is nearly 600 times larger, so it seemed unusual that the reconstruction would run almost twice as fast as for the smaller volume. Please clarify.

Response:

While being significantly smaller, the 32×32 array has dense histograms of 153 bins with non-zero counts, due to the accumulation of 3008 binary frames per lidar frame (please see comment 2.5). On the other hand, the 141×141 raster scan dataset has a mean photon count of 3 photons per pixel, hence having approximately 3 active bins per pixel. Hence, the effective data size in the former case is $32 \times 32 \times 153 = 156672$, whereas in the latter is $141 \times 141 \times 3 \times 2 = 119286$ (where the last term in the multiplication is due to the bin number indicator in a sparse representation). The latter data size is smaller than the 32×32 array, hence the faster processing. Moreover, as the algorithm's complexity is driven by the amount of computation within a pixel, it is more intensive to process 153 bins than 4 active bins. We have included an example illustrating this phenomenon in the updated supplementary material.

Comment 2.5

The authors report that the 2D SPAD array captures 150,400 binary frames per second. Does this mean that the sensor provides a maximum output rate of 150k histograms per second? Is there a limit of how many photon timestamps can be captured per histogram?

Response:

Each binary frame contains at most one time-tagged photon detection per pixel position, thus it is indeed possible to obtain 150k histograms per second, but only having at most one photon per histogram. In our experiments, we accumulated the photon detections into 50 histograms per second, that can have at most 3008 photon detections per histogram (i.e., 3008 binary frames). This explanation has been included in the updated paper (section '3D Dynamic scenes results').

Comment 2.6

I wasn't entirely convinced about the benefit of the upsampling from 32×32 spatial resolution to 96×96 demonstrated in the dynamic results. Do the super-resolved results show much improvement over a naive upscaling of a native 32×32 reconstruction?

Response:

We have sought to clarify the benefits of upsampling. We believe that the proposed upsampling brings additional details to the reconstructed objects, improving the estimates of naive upsampling in a post-processing step. For example, the figure below shows the upsampled reconstructions with the proposed method and cross-correlation. The cross-correlation output was upsampled by naively converting each detection into a 3×3 grid of points at the same depth. While the upsampled cross-correlation has a blocky appearance, the proposed method captures additional details in the contours of the 3D target. Note that these contours are not always aligned with the coarse scale.

Comparison of 3D reconstructions of the 32×32 lidar array data using cross-correlation and the proposed method. The upsampling strategy of the proposed method brings additional details in the contours of the object, whereas a naive upsampling of the cross-correlation output presents a blocky appearance.

This short discussion has been included in the section 'Improvements by upsampling in small lidar arrays' of the updated supplementary material.

Comment 2.7

For Fig. 5 of the methods section, please clarify how many active bins are used in the execution time vs LIDAR pixel comparison and how many LIDAR pixels are used in the execution time vs active bins comparison.

Response:

In Fig. 5 (a), the execution time was reported for a dataset with a mean of 4 active bins per pixel, whereas Fig. 5 (b) was generated for an array of 141×141 pixels. This information has been included in the caption of Fig. 5 of the updated manuscript.

References

- [1] F. Heide, S. Diamond, D. B. Lindell, and G. Wetzstein, “Sub-picosecond photon-efficient 3D imaging using single-photon sensors,” *Scientific reports*, vol. 8, no. 1, p. 17726, 2018.
- [2] A. Kirmani, D. Venkatraman, D. Shin, A. Colaço, F. N. Wong, J. H. Shapiro, and V. K. Goyal, “First-photon imaging,” *Science*, vol. 343, no. 6166, pp. 58–61, 2014.
- [3] D. Shin, A. Kirmani, V. K. Goyal, and J. H. Shapiro, “Photon-efficient computational 3-D and reflectivity imaging with single-photon detectors,” *IEEE Transactions on Computational Imaging*, vol. 1, no. 2, pp. 112–125, 2015.
- [4] A. Halimi, Y. Altmann, A. McCarthy, X. Ren, R. Tobin, G. S. Buller, and S. McLaughlin, “Restoration of intensity and depth images constructed using sparse single-photon data,” in *Proc. 24th European Signal Processing Conference (EUSIPCO), Budapest, Hungary, Aug. 2016*, pp. 86–90.
- [5] J. Rapp and V. K. Goyal, “A few photons among many: Unmixing signal and noise for photon-efficient active imaging,” *IEEE Transactions on Computational Imaging*, vol. 3, no. 3, pp. 445–459, 2017.
- [6] J. Bolte, S. Sabach, and M. Teboulle, “Proximal alternating linearized minimization for nonconvex and nonsmooth problems,” *Mathematical Programming*, vol. 146, no. 1, pp. 459–494, Aug 2014. [Online]. Available: <https://doi.org/10.1007/s10107-013-0701-9>
- [7] J. Tachella, Y. Altmann, X. Ren, A. McCarthy, G. Buller, S. McLaughlin, and J. Tourneret, “Bayesian 3D reconstruction of complex scenes from single-photon lidar data,” *SIAM Journal on Imaging Sciences*, vol. 12, no. 1, pp. 521–550, 2019. [Online]. Available: <https://doi.org/10.1137/18M1183972>
- [8] D. Shin, F. Xu, F. N. Wong, J. H. Shapiro, and V. K. Goyal, “Computational multi-depth single-photon imaging,” *Optics express*, vol. 24, no. 3, pp. 1873–1888, 2016.
- [9] A. Halimi, R. Tobin, A. McCarthy, S. McLaughlin, and G. S. Buller, “Restoration of multi-layered single-photon 3d lidar images,” in *Proc. 25th European Signal Processing Conference (EUSIPCO), Kos Island, Greece, Aug. 2017*, pp. 708–712.
- [10] J. Tachella, Y. Altmann, M. Márquez, H. Arguello-Fuentes, J.-Y. Tourneret, and S. McLaughlin, “Bayesian 3D Reconstruction of Subsampled Multispectral Single-photon Lidar Signals,” *arXiv e-prints*, p. arXiv:1904.02583, Apr 2019.
- [11] V. Chandrasekaran and M. I. Jordan, “Computational and statistical tradeoffs via convex relaxation,” *In Proc. of the National Academy of Sciences*, vol. 110, no. 13, pp. E1181–E1190, 2013. [Online]. Available: <https://www.pnas.org/content/110/13/E1181>
- [12] M. Berger, A. Tagliasacchi, L. M. Seversky, P. Alliez, G. Guennebaud, J. A. Levine, A. Sharf, and C. T. Silva, “A survey of surface reconstruction from point clouds,” *Computer Graphics Forum*, vol. 36, no. 1, pp. 301–329, 2017. [Online]. Available: <https://onlinelibrary.wiley.com/doi/abs/10.1111/cgf.12802>
- [13] S. Xiong, J. Zhang, J. Zheng, J. Cai, and L. Liu, “Robust surface reconstruction via dictionary learning,” *ACM Transactions on Graphics (TOG)*, vol. 33, no. 6, p. 201, 2014.
- [14] C.-H. Shen, S.-S. Huang, H. Fu, and S.-M. Hu, “Adaptive partitioning of urban facades,” in *ACM Transactions on Graphics (TOG)*, vol. 30, no. 6. ACM, 2011, p. 184.

- [15] L. Nan, K. Xie, and A. Sharf, “A search-classify approach for cluttered indoor scene understanding,” *ACM Trans. Graph.*, vol. 31, no. 6, pp. 137:1–137:10, Nov. 2012. [Online]. Available: <http://doi.acm.org/10.1145/2366145.2366156>
- [16] M. M. Bronstein, J. Bruna, Y. LeCun, A. Szlam, and P. Vandergheynst, “Geometric deep learning: Going beyond euclidean data,” *IEEE Signal Processing Magazine*, vol. 34, no. 4, pp. 18–42, July 2017.
- [17] Y. Li, R. Bu, M. Sun, W. Wu, X. Di, and B. Chen, “PointCNN: Convolution on X-transformed points,” in *Advances in Neural Information Processing Systems (NIPS 2018)*, Montreal, Canada. Curran Associates, Inc., 2018, pp. 820–830. [Online]. Available: <http://papers.nips.cc/paper/7362-pointcnn-convolution-on-x-transformed-points.pdf>
- [18] Y. Wang, Y. Sun, Z. Liu, S. E. Sarma, M. M. Bronstein, and J. M. Solomon, “Dynamic Graph CNN for Learning on Point Clouds,” *arXiv e-prints*, p. arXiv:1801.07829, Jan 2018.

Reviewers' Comments:

Reviewer #1:

Remarks to the Author:

I am pleased with the revision and have no additional suggestions.

Vivek Goyal

Reviewer #2:

Remarks to the Author:

I appreciate the authors detailed attention to the reviewers' comments and their thoughtful responses. After reviewing the point-by-point response and the updated manuscript, I feel that the authors adequately addressed my comments from the previous review round.